# The SARS-CoV-2 Alpha variant was associated with increased clinical severity of COVID-19 in Scotland: A genomics-based retrospective cohort analysis

**David J. Pascall**[1,2�)], **Elen Vink**[3,4☺], **Rachel Blacow**[3,5], **Naomi Bulteel**[6], **Alasdair Campbell**[4], **Robyn Campbell**[4], **Sarah Clifford**[4], **Chris Davis**[3], **Ana da Silva Filipe**[3], **Noha El Sakka**[7], **Ludmila Fjodorova**[5], **Ruth Forrest**[6], **Emily Goldstein**[5], **Rory Gunson**[5], **John Haughney**[5], **Matthew T. G. Holden**[8,9], **Patrick Honour**[10], **Joseph Hughes**[3], **Edward James**[10], **Tim Lewis**[4], **Samantha Lycett**[11], **Oscar MacLean**[3], **Martin McHugh**[4], **Guy Mollett**[3,5], **Yusuke Onishi**[7], **Ben Parcell**[12], **Surajit Ray**[13], **David L. Robertson**[3], **Sharif Shaaban**[8], **James G. Shepherd**[3], **Katherine Smollett**[3], **Kate Templeton**[4], **Elizabeth Wastnedge**[4], **Craig Wilkie**[13], **Thomas Williams**[4,14], **Emma C. Thomson**[3,5,15]*, **The COVID-19 Genomics UK (COG-UK) Consortium**[¶]

1 MRC Biostatistics Unit, University of Cambridge, Cambridge, United Kingdom, 2 Joint Universities Pandemic and Epidemiological Research (JUNIPER) Consortium, United Kingdom, 3 MRC–University of Glasgow Centre for Virus Research (CVR), Glasgow, United Kingdom, 4 NHS Lothian, Edinburgh, United Kingdom, 5 NHS Greater Glasgow and Clyde, Glasgow, United Kingdom, 6 NHS Fife, Kirkcaldy, Fife, United Kingdom, 7 NHS Grampian, Aberdeen, United Kingdom, 8 Public Health Scotland, Edinburgh, United Kingdom, 9 School of Medicine, University of St Andrews, St Andrews, Fife, United Kingdom, 10 NHS Borders, Melrose, Scottish Borders, United Kingdom, 11 The Roslin Institute, University of Edinburgh, Midlothian, United Kingdom, 12 School of Life Sciences, University of Dundee, Dundee, United Kingdom, 13 School of Mathematics and Statistics, University of Glasgow, Glasgow, United Kingdom, 14 Royal Hospital for Children and Young People, University of Edinburgh, Edinburgh, United Kingdom, 15 London School of Hygiene and Tropical Medicine, London, United Kingdom

☺ These authors contributed equally to this work.
¶ Membership of the COVID-19 Genomics UK (COG-UK) Consortium is provided in the Acknowledgments section.
* emma.thomson@glasgow.ac.uk

**Data Availability Statement:** Data cannot be shared publicly due to ethical constraints on data sharing because it contains sensitive patient data.

## Abstract

### Objectives

The SARS-CoV-2 Alpha variant was associated with increased transmission relative to other variants present at the time of its emergence and several studies have shown an association between Alpha variant infection and increased hospitalisation and 28-day mortality. However, none have addressed the impact on maximum severity of illness in the general population classified by the level of respiratory support required, or death. We aimed to do this.

### Methods

In this retrospective multi-centre clinical cohort sub-study of the COG-UK consortium, 1475 samples from Scottish hospitalised and community cases collected between 1st November 2020 and 30th January 2021 were sequenced. We matched sequence data to clinical outcomes as the Alpha variant became dominant in Scotland and modelled the association

As such, full data cannot be removed from the NHS GG&C SafeHaven, a trusted research environment without appropriate permissions. Aggregated data are available in the supplementary materials. The Safe Haven can be contacted for data access requests safehaven@ggc.scot.nhs.uk.

**Funding:** COG-UK is supported by funding from the Medical Research Council (MRC) part of UK Research & Innovation (UKRI), the National Institute of Health Research (NIHR) and Genome Research Limited, operating as the Wellcome Sanger Institute. Funding was also provided by UKRI through the JUNIPER consortium (MR/V038613/1). Sequencing, bioinformatics and statistical support was funded by the Medical Research Council (MRC) core awards for the MRC-University of Glasgow Centre for Virus Research (MC UU 1201412) and MRC Biostatistics Unit (MC UU 00002/11). The funders had no role in study design, data collection and analysis, decision to publish, or preparation of the manuscript.

**Competing interests:** The authors have declared that no competing interests exist.

between Alpha variant infection and severe disease using a 4-point scale of maximum severity by 28 days: 1. no respiratory support, 2. supplemental oxygen, 3. ventilation and 4. death.

## Results

Our cumulative generalised linear mixed model analyses found evidence (cumulative odds ratio: 1.40, 95% CI: 1.02, 1.93) of a positive association between increased clinical severity and lineage (Alpha variant versus pre-Alpha variants).

## Conclusions

The Alpha variant was associated with more severe clinical disease in the Scottish population than co-circulating lineages.

## Introduction

The Alpha variant of SARS-CoV-2 (Pango lineage B.1.1.7) was first identified in the UK in September 2020 and was subsequently reported in 183 countries [1]. It is defined by 21 genomic mutations or deletions, including 8 characteristic changes within the spike gene (S1 Table) [2]. These are associated with increased ACE-2 receptor binding affinity and innate and adaptive immune evasion [3–6] compared to preceding lineages. The Alpha variant, the first variant of concern (VOC), was estimated to be 50–100% more transmissible than other lineages present at the time of its emergence [7], explaining the transient dominance of this lineage globally.

The presence of a spike gene deletion (Δ69–70) results in spike-gene target failure (SGTF) in real-time reverse transcriptase polymerase chain reaction (RT-PCR) diagnostic assays and provided a useful proxy for the presence of the Alpha variant for epidemiological analysis during this time period [2]. Four large community analyses showed a positive association between the presence of SGTF and 28-day mortality, with hazard ratios of 1.55 (CI 1.39–1.72), 1.64 (CI 1.32–2.04), 1.67 (CI 1.34–2.09) and 1.73 (CI 1.41–2.13) [8–10]. Both SGTF (hazard ratios of 1.52 (CI 1.47–1.57), 1.62 (CI 1.48–1.78)) [11, 12] and confirmed Alpha variant infection (hazard ratios of 1.34 (CI 1.07–1.66) and 1.61 (CI 1.28–2.03) [12, 13] were associated with an increased risk of hospitalisation in community cases, and a smaller study of hospitalised patients found a greater risk of hypoxia at admission in those with confirmed Alpha variant infection [14]. In contrast, other smaller analyses of hospitalised patients found no association between confirmed Alpha variant infection and increased clinical severity based on a variety of indices [15–17]. Limited data are available on the full clinical course of disease with the Alpha variant in relation to co-circulating variants.

Understanding the clinical pattern of disease with new variants of concern is important for several reasons. Firstly, if a variant is more pathogenic than previous variants, this has implications for considering public health restrictions and the optimal functioning of health care systems. Secondly, large numbers of low- and middle-income countries still have less than 50% of their populations having been vaccinated against SARS-CoV-2 [18]. A better understanding of a variant with increased severity is important in modelling the impact of unmitigated infection in these settings. A clear understanding of the behaviour of the Alpha variant, which emerged as a dominant variant in Scotland in the winter of 2020/21, is needed as a baseline to compare the clinical phenotype of variants of concern that have subsequently emerged. Post-Alpha

variants, such as Omicron (B.1.1.529), have been shown to be able to evade vaccine-induced immunity and therefore have the potential to spread even in immunised populations [19], so a historical understanding of severity remains important, as it seems unlikely that SARS-CoV-2 infections will be brought under control in the near future.

We aimed to quantify the clinical features and rate of spread of Alpha variant infections in Scotland in a comprehensive national dataset. We used whole genome sequencing data to analyse patient presentations between 1st November 2020 and 30th January 2021 as the variant emerged in Scotland and used cumulative generalised additive models to compare 28-day maximum clinical severity for the Alpha variant against co-circulating lineages.

## Materials and methods

### Sample collection and approvals

We included all Scottish COG-UK pillar 1 samples sequenced at the MRC-University of Glasgow Centre for Virus Research (CVR) and the Royal Infirmary of Edinburgh (RIE) between 1st November 2020 and 30th January 2021. These samples derived from both hospitalised patients (59%) and community testing (41%).

Residual nucleic acid extracts derived from the nose-throat swabs of SARS-CoV-2 positive individuals whose diagnostic samples were submitted to the West of Scotland Specialist Virology Centre and Edinburgh Royal Infirmary Virus laboratory and were sequenced following ethical approvals from the West of Scotland Biorepository (16/WS/0207NHS) and the Lothian Biorepository (10/S1402/33). These samples were sequenced without consent following HTA legislation on consent exemption. Use of Scottish anonymised clinical data linked to virus genomic data without informed consent was granted by the Caldicott guardian for each site and by the Scottish Public Benefit and Privacy Panel (PBPP) for Health and Social Care (2122–0130).

### Sequencing and bioinformatics

Sequencing was performed as part of the COG-UK consortium using amplicon-based next generation sequencing [20, 21]. Sequence alignment, lineage assignment and tree generation were performed using the COG-UK data pipeline (https://github.com/COG-UK/datapipe) and phylogenetic pipeline (https://github.com/cov-ert/phylopipe) with pangolin lineage assignment (https://github.com/cov-lineages/pangolin) [22]. Lineage assignments were performed on 18/03/2021 and phylogenetic analysis was performed using the COG-UK tree generated on 25/02/2021. Estimates of growth rates of major lineages in Scotland were calculated from time-resolved phylogenies for lineages B.1.1.7 (Alpha), B.177 and the sub-clades B.177.5, B.177.8, and another minor B.177 sub-clade (W.4). The estimates were carried out utilising sequences from November 2020 –March 2021 in BEAST (Bayesian Evolutionary Analysis by Sampling Trees) with an exponential growth rate population model, strict molecular clock model and TN93 with four gamma rate distribution categories. Each lineage was randomly subsampled to a maximum of 5 sequences per epiweek (resulting in 52 to 103 sequences per subsample, depending on the lineage), and 10 subsamples replicates analysed per lineage in a joint exponential growth rate population model.

### Clinical data

Core demographic data (age, sex, partial postcode) were collected via linkage to electronic patient records at the 7 of 14 scottish health boards (covering 78% of the scottish population) for which we had clinical data access approval, and a full retrospective review of case notes was

undertaken. Collected data included residence in a care home; occupation in care home or healthcare setting; admission to hospital; date of admission, discharge and/or death and maximum clinical severity at 28 days sample collection date via a 4-point ordinal scale (1. No respiratory support; 2. Supplemental low flow oxygen; 3. Invasive ventilation, non-invasive ventilation or high-flow nasal canula (IV/NIV/HFNO); 4. Death) as previously used in Volz et al 2020 and Thomson et al 2021 [23, 24].

Where available, PCR (Polymerase Chain Reaction) cycle threshold (Ct) and the PCR testing platform were recorded. Nosocomial COVID-19 was defined as a first positive PCR occurring greater than 48 hours following admission to hospital, individuals meeting this criterion were excluded from the study. Discharge status was followed up until 15th April 2021 for the hospital stay analysis. For the co-morbidity sub-analysis, delegated research ethics approval was granted for linkage to National Health Service (NHS) patient data by the Local Privacy and Advisory Committee at NHS Greater Glasgow and Clyde. Cohorts and de-identified linked data were prepared by the West of Scotland Safe Haven at NHS Greater Glasgow and Clyde.

## Severity analyses

Four level severity data was analysed using cumulative (per the definition of Bürkner and Vuorre (2019)) generalised additive mixed models (GAMMs) with logit links, specifically, following Volz et al (2020) [23, 25]. We analysed three subsets of the data: 1. the full dataset, 2. the dataset excluding care home patients, and 3. exclusively the hospitalised population. Further details regarding these analyses are provided in S1 Appendix.

## Ct analysis

Ct value was compared between Alpha variant and pre-Alpha variant infections for those patients where the TaqPath assay (Applied Biosystems) was used. This platform was used exclusively for this analysis because different platforms output systematically different Ct values, and this was the most frequently used in our dataset (n = 154, Alpha = 38, pre-Alpha = 116). We used a generalised additive model with a Gaussian error structure and identity link, and the same covariates used as in the severity analysis to model the Ct value. The model was fitted using the brms (v. 2.14.4) R package [26]. The presented model had no divergent transitions and effective sample sizes of over 200 for all parameters. The intercept of the model was given a t-distribution (location = 20, scale = 10, df = 3) prior, the fixed effect coefficients were given normal (mean = 0, standard deviation = 5) priors, random effects and spline standard deviations were given exponential (mean = 5) priors.

## Hospital length of stay analysis

Hospital length of stay was compared for Alpha variant and pre-Alpha variant patients while controlling for age and sex using a Fine and Gray model competing risks regression using the crr function in the cmprsk (v. 2.2–10) R package [27, 28]. Nosocomial infections were excluded. In total, this analysis had 521 cases (Alpha = 187, pre-Alpha = 334), of which 4 were censored; 352 patients were discharged from hospital and 165 died.

## Results

## Emergence of the Alpha variant in Scotland

Between 01/11/2020 and 31/01/2021 1863 samples from individuals tested in pillar 1 facilities in Scotland underwent whole genome sequencing for SARS-CoV-2. Of these, 1475 (79%)

could be linked to patient records from participating scottish health boards, and were included in the analysis. The contribution of patients infected with the Alpha variant increased over the course of the study, in line with dissemination across the UK during the study period (Fig 1A and 1B). At the time of data collection, two peaks of SARS-CoV-2 infection had occurred in the UK: the first (wave 1) in March 2020 [15] and the second in summer 2020 [29], both in association with hundreds of importations following travel to Central Europe [30]. The second peak incorporated two variant waves (waves 2 and 3), initially of B.1.177 (Fig 1C) and then B.1.1.7/Alpha, radiating from the South of England (Fig 1E). This Alpha variant "takeover" (Fig 1D) corresponded to a five-fold increase in growth rate on an epidemiological scale relative to pre-Alpha lineages (Fig 1F).

## Demographics of the clinical cohort

The age of the clinical cohort ranged from 0–105 years, (mean 66.8 years) and was slightly lower in the Alpha group (65.6 years vs. 67.2 years). Overall, 59.1% were female; this preponderance occurred in both subgroups and was higher in the Alpha subgroup (60.4% vs 58.6%). In the full cohort, 3.0% were care-home workers and 10.4% were NHS healthcare workers. 5.5% and 5.8% of those infected with the Alpha variant were care-home and other healthcare workers respectively, compared with 2.2% and 12.0% of those infected with pre-Alpha lineages. 12.9% of those in the Alpha subgroup were care-home residents, compared with 21.7% in pre-Alpha. There was also a difference in the proportion of cases admitted to Intensive Care Units: 6.3% of the Alpha group compared with 3.4% for pre-Alpha. Full details of the demographic data of the cohort can be found in Table 1 and full lineage assignments can be found in S2 Table.

## Clinical severity analysis

Within the clinical severity cohort there were 364 Alpha cases, 1030 B.1.177 cases, and 81 cases due to one of 19 other pre-Alpha lineages (Fig 2), of which 185 Alpha cases (51%) and 440 pre-Alpha cases (38%) received oxygen or died. Consistent with previous research comparing mortality and hospitalisation in SGTF detected by PCR versus absence of SGTF, we found that Alpha variant viruses were associated with more severe disease on average than those from other lineages circulating during the same time period. In the full dataset, we observed a positive association with severity (posterior median cumulative odds ratio: 1.40, 95% CI: 1.02–1.93). In both the subsets, excluding care-home patients or limiting to hospitalised patients only, the mean estimate of the increase in severity of the Alpha variant was smaller, and the variance in the posterior distribution higher likely due to the smaller sample sizes. Given this uncertainty, we cannot determine whether the association of the Alpha variant with severity in the populations corresponding to these subsets is the same as that in the population described by the full dataset, but in all cases, the most likely direction of the effect is positive. Comorbidity data were not available for the full dataset, a sub-analysis on those cases where it could be linked indicated that comorbidities did not substantially affect relative severity estimates (S3 Appendix). Model estimates from severity models from all subsets can be found in S3–S5 Tables.

Bernoulli models looking at sequential severity categories provided weak evidence that the proportional odds assumption of the cumulative logistic model was violated. The odds ratios for the no oxygen versus low flow oxygen, and low flow oxygen versus IV/NIV/HFNC were similar to those estimated under the cumulative model (posterior median odds ratio for no oxygen versus low flow oxygen: 1.77, CI: 1.12–2.80; posterior median odds ratio for low flow oxygen versus IV/NIV/HFNC: 1.26, CI: 0.43–3.67) but with correspondingly higher posterior

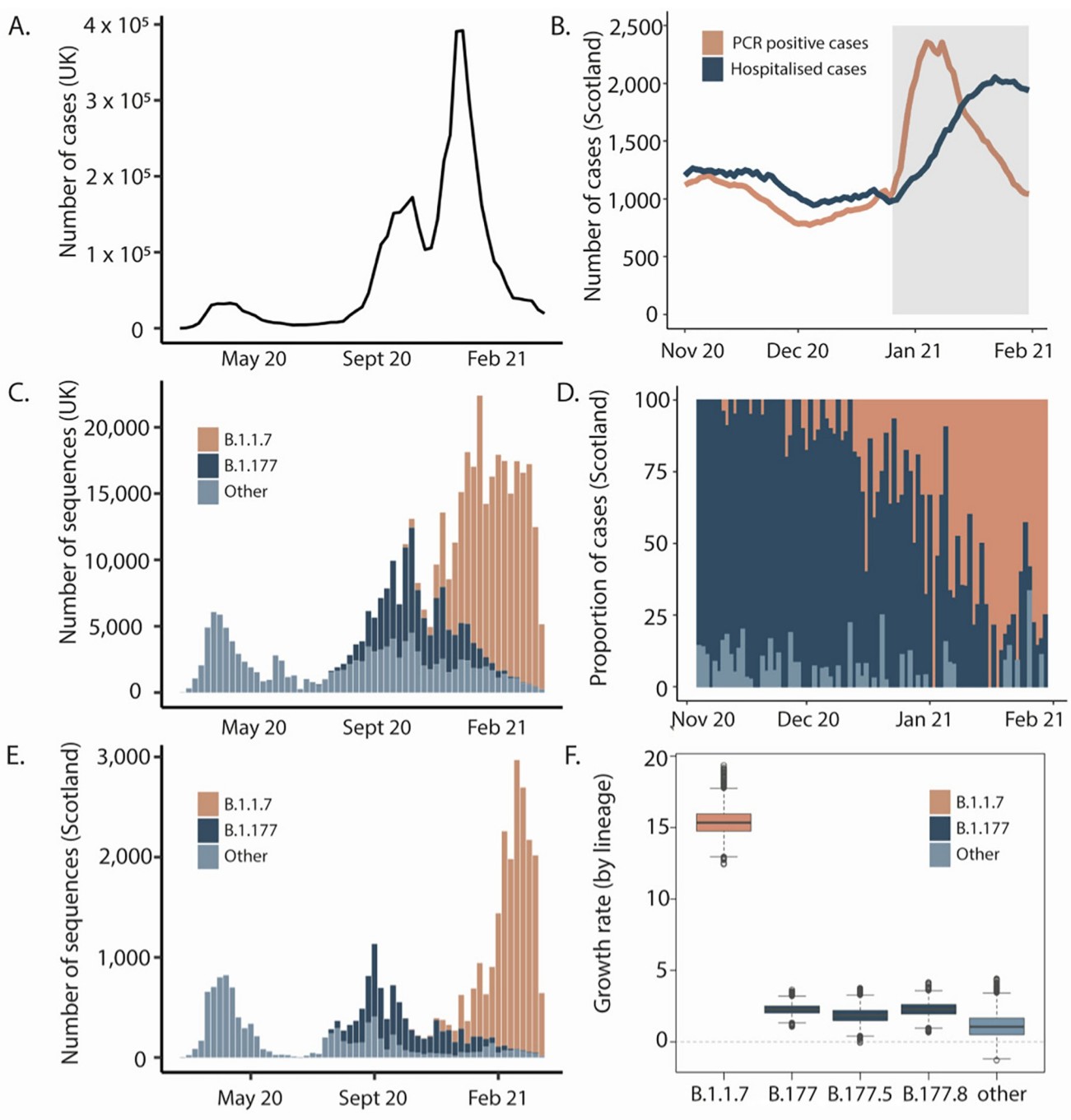

**Fig 1. Introduction and growth of the Alpha variant (lineage B.1.1.7) in the UK, 2020/21.** A) Waves of SARS-CoV-2 confirmed cases in the UK. B) Seven-day rolling average of daily PCR positive cases (orange) and total number of patients hospitalised (dark blue) with COVID-19 in Scotland during the study period. Grey shaded area represents the period of lockdown beginning 26/12/2020. C) Variants in the UK. D) Proportion of cases by lineage in the clinical severity cohort. E) Variants in Scotland showing three distinct waves in winter and early spring 2020, summer 2020 and autumn/winter, attributed to the shifts from B1 and other variants (light blue) to B.1.177 (dark blue) and then B.1.1.7/Alpha (orange). Waves one and two closely mirror the broader UK situation as they are linked to both continental European and introductions from England. Wave three has a single origin in Kent so Scotland lags England in numbers of cases. F) Estimates of growth rates of major lineages in Scotland from time-resolved phylogenies. Estimates were carried out on a subsample of the named lineages using sequences from Scotland only from November 2020-March 2021 using BEAST and an exponential growth effective population size model.

**Table 1. Demographic characteristics of Scottish patients infected with SARS-CoV-2 by lineage.**

| Characteristic | Overall Group (n = 1475) | | B.1.1.7 (Alpha) (n = 364) | | Other (Pre-Alpha) (n = 1111) | |
|---|---|---|---|---|---|---|
| | Number | Percentage | Number | Percentage | Number | Percentage |
| Age at diagnosis (years) | | | | | | |
| Mean ± SD | 66.8±20.8 | | 65.6±20.6 | | 67.2±20.8 | |
| Range | 0–105 | | 0–105 | | 0–100 | |
| Sex | | | | | | |
| Male | 604 | 40.9% | 144 | 39.6% | 460 | 41.4% |
| Female | 871 | 59.1% | 220 | 60.4% | 651 | 58.6% |
| Admitted to hospital | | | | | | |
| Yes | 876 | 59.4% | 238 | 65.4% | 638 | 57.4% |
| No | 599 | 40.6% | 126 | 34.6% | 473 | 42.6% |
| Care home worker | | | | | | |
| Yes | 44 | 3.0% | 20 | 5.5% | 24 | 2.2% |
| No | 1305 | 88.5% | 305 | 83.8% | 1000 | 90.0% |
| Unknown | 126 | 8.5% | 39 | 10.7% | 87 | 7.8% |
| Non-care home healthcare worker | | | | | | |
| Yes | 154 | 10.4% | 21 | 5.8% | 133 | 12.0% |
| No | 1193 | 80.9% | 305 | 83.8% | 888 | 79.9% |
| Unknown | 128 | 8.7% | 38 | 10.4% | 90 | 8.1% |
| Nursing home resident | | | | | | |
| Yes | 288 | 19.5% | 47 | 12.9% | 241 | 21.7% |
| No | 1187 | 80.5% | 317 | 87.1% | 870 | 78.3% |
| Unknown | 0 | 0.0% | 0 | 0.0% | 0 | 0.0% |
| Diagnosis >48 hours post-admission | | | | | | |
| Yes | 346 | 23.5% | 46 | 12.6% | 300 | 27.0% |
| No | 1040 | 70.5% | 289 | 79.4% | 751 | 67.6% |
| Unknown | 89 | 6.0% | 29 | 8.0% | 60 | 5.4% |
| Travel outside Scotland | | | | | | |
| Yes | 1 | 0.1% | 0 | 0.0% | 1 | 0.1% |
| No | 317 | 21.5% | 20 | 5.5% | 297 | 26.7% |
| Unknown | 1157 | 78.4% | 302 | 94.5% | 813 | 73.2% |
| Immunosuppressed | | | | | | |
| Yes | 42 | 2.9% | 4 | 1.1% | 38 | 3.4% |
| No | 474 | 31.1% | 60 | 16.5% | 414 | 37.3% |
| Unknown | 959 | 65.0% | 300 | 82.4% | 659 | 59.3% |
| Visited Intensive Care Unit? | | | | | | |
| Yes | 61 | 4.1% | 23 | 6.3% | 38 | 3.4% |
| No | 1413 | 95.8% | 341 | 93.7% | 1072 | 96.5% |
| Unknown | 1 | 0.1% | 0 | 0.0% | 1 | 0.1% |
| Patient alive/deceased? | | | | | | |
| Alive | 1115 | 75.6% | 273 | 75.0% | 842 | 75.8% |
| Deceased | 360 | 24.4% | 91 | 25.0% | 269 | 24.2% |

variances given the smaller sample size. The odds ratios for the IV/NIV/HFNC versus death model suggested that the preponderance of evidence was in favour of Alpha infection associated with lower risk of death, conditional on having received IV, NIV or HFNC (posterior median odds ratio: 0.64, CI: 0.22–1.90). However, the credible intervals here are wide, given the sample size, and do include the estimated global effect. A similar but more extreme effect

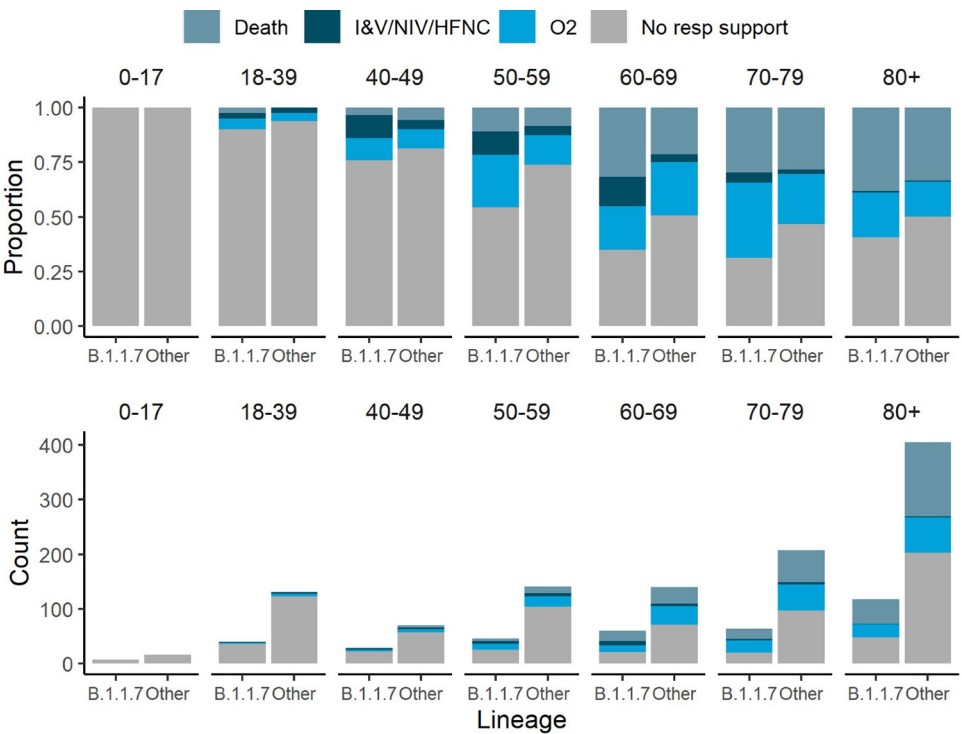

**Fig 2. Comparison of disease severity between the Alpha variant (B.1.1.7) and other lineages.** Clinical severity was measured on a four-level ordinal scale based on the level of respiratory support received for 1454 patients stratified by age group; death, invasive or non-invasive ventilatory support including high flow nasal cannulae (IV/NIV/HFNC), supplemental oxygen delivered by low flow mask devices or nasal cannulae, and no respiratory support.

was observed for the effect of biological sex, with male sex being associated with more severe outcomes for the first two sequential category models (posterior median odds ratio for no oxygen versus low flow oxygen: 1.32, CI: 0.96–1.80; posterior median odds ratio for low flow oxygen versus IV/NIV/HFNC: 3.10, CI: 1.37–7.08), but withless severe outcomes for the last (posterior odds ratio for IV/NIV/HFNC vs death: 0.62, CI: 0.19–099). Given other research on the topic has consistently identified male sex as a risk factor, this potentially indicates the existence of an important unmeasured confounder only relevant for those requiring invasive ventilation, non-invasive ventilation or high flow nasal cannula oxygen.

Estimates of the severity across the phylogeny are visible in Fig 3; see S2 Appendix for more discussion of this analysis. An analysis including comorbidities for the subset of patients where they were available implied that the inclusion of comorbidities had no impact on the results obtained, see S1 and S3 Appendices.

We also found that the Alpha variant was associated with lower Ct values than infection with pre-Alpha variants (posterior median Ct change: -2.46, 95% CI: -4.22 - -0.70) as previously observed [8]. Model estimates for all parameters can be found in S6 Table.

We found no evidence that the Alpha variant was associated with longer hospital stays after controlling for age and sex (HR: -0.02; 95% CI: -0.23–0.20; p = 0.89).

## Discussion

In this analysis of hospitalised and community patients with Alpha variant and pre-Alpha variant SARS-CoV-2 infection, carried out as the Alpha variant became dominant in Scotland, we

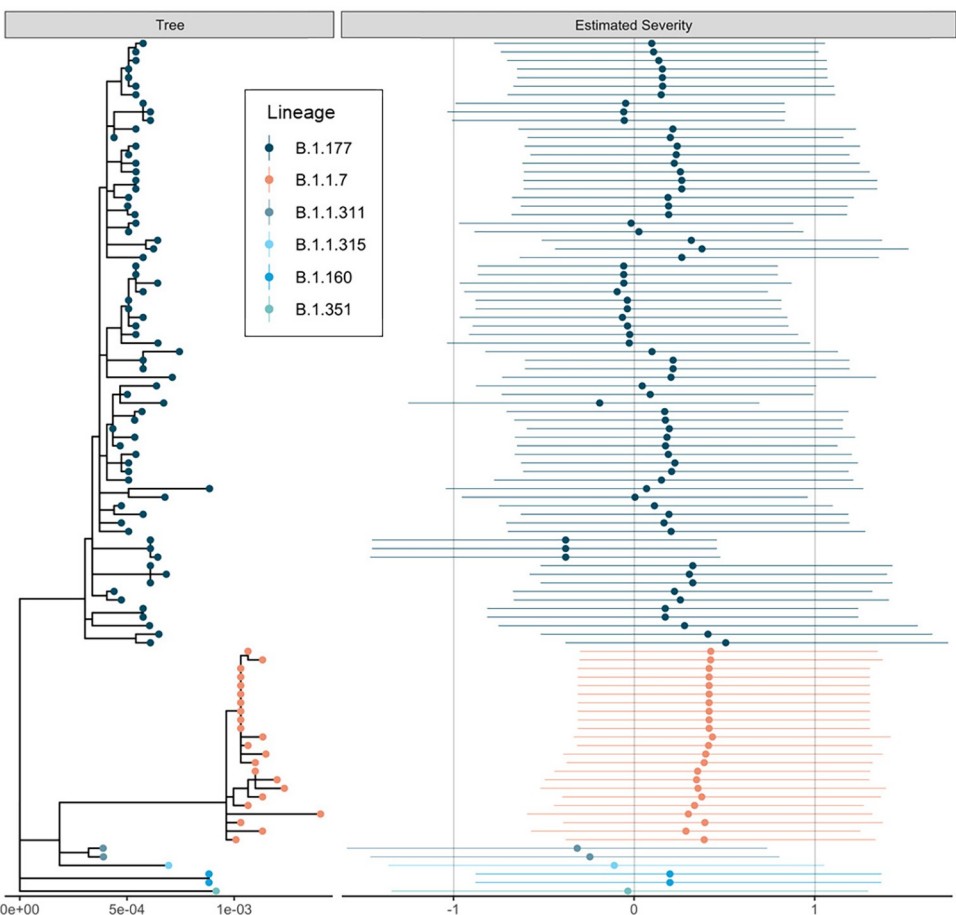

**Fig 3. The estimated maximum likelihood phylogenetic tree and a measure of estimated severities of infection.**
Estimated severities for each viral isolate are means and 95% credible intervals of the linear predictor change under infection with that viral genotype from the phylogenetic random effect in the cumulative severity model under a Brownian motion model of evolution. This model constrains genetically identical isolates to have identical effects, so changes should be interpreted across the phylogeny rather than between closely related isolates which necessarily have similar estimated severities. The dataset was downsampled to 100 random samples for this figure to aid readability. Figure was generated using ggtree [31].

provide evidence of increased clinical severity associated with this variant at this time, after adjusting for age, sex, geography and calendar time, as well as testing for sensitivity to number of comorbidities. This was observed across all adult age groups, incorporating the spectrum of COVID-19 disease; from no requirement for supportive care, to supplemental oxygen requirement, the need for invasive or non-invasive ventilation, and to death. This analysis is the first to assess the full clinical severity spectrum of confirmed Alpha variant infection in both community and hospitalised cases in relation to other prevalent lineages circulating during the same time period.

Our study supports the community testing analyses that have reported an increased 28-day mortality associated with SGTF as a proxy for Alpha variant status [8–10]. Smaller studies found no effect of lineage on various measures of severity [15–17], but these were studies of patients already admitted to hospital and therefore would not pick up the granular detail of increasing disease severity resulting in a need for increasing levels of respiratory support and consequently admission to hospital.

The association between higher viral load, higher transmission and lineage may reflect changes in the biology of the virus; for example, the Alpha variant asparagine (N) to tyrosine (Y) mutation at position 501 of the spike protein receptor binding domain (RBD) was associated with an increase in binding affinity to the human ACE2 receptor [32]. In addition, a deletion at position 69–70 may have increased virus infectivity [33]. The P681H mutation found at the furin cleavage site is associated with more efficient furin cleavage, enhancing cell entry [34]. An alternative explanation for the higher viral loads observed in Alpha variant infection may be that clinical presentation occurs earlier in the illness. Further modelling, animal experiments and studies in healthy volunteers may help to unravel the mechanisms behind this phenomenon.

Our data indicate an association between the Alpha variant and an increased risk of requiring supplemental oxygen and ventilation compared to per-Alpha variants. These two factors are critical determinants of healthcare capacity during a period of high incidence of SARS-CoV-2 infection, and this illustrates the importance for countries, in particular those with less robust health care systems and lower vaccination rates of factoring the requirement for supportive treatment into models of clinical severity and pandemic response decision planning for future SARS-CoV-2 variants of concern. This granular analysis of disease severity based on genomic confirmation of diagnosis should be used as a baseline study for clinical severity analysis of the inevitable future variants of concern.

There are some limitations to our study. Our dataset is drawn from first-line local NHS diagnostic (Pillar 1) testing which over-represents patients presenting for hospital care (59%) while those sampled in the community represented 41% of the dataset. The effect of working in the healthcare sector on severity, driven by systematically different exposures faced by frontline caregivers, could not be adjusted for, due to incompleteness of the data regarding this variable. Further, the analysis dataset employed a non-standardised approach to sampling across the study period as sequencing was carried out both as systematic randomised national surveillance and sampling following outbreaks of interest. Additionally, we did not have information about the vaccination status of the individuals in the study. However, our inability to adjust for this variable is not likely to have had a great impact on our conclusions, as, at the time of the study, the vaccination campaign had recently begun, with only over 75 year olds and high-risk groups eligible. Finally, the cumulative model used and the usage of a single (not location varying) spine for the effect of time in this analysis assumes a homogenous application of therapeutic intervention across the population. Despite these limitations, our results remain consistent with previous work on the mortality of Alpha, and this study provides new information regarding differences in infection severity.

In summary, the Alpha variant was found to be associated with a rapid increase in COVID-19 cases in Scotland in the winter of 2020/21, and an increased risk of severe infection requiring supportive care. This has implications for planning for future variant driven waves of infection, especially in countries with low vaccine uptake or if variants evolve with significant vaccine-escape. Our study has shown the value of the collection of higher resolution patient outcome data linked to genetic sequences when looking for clinically relevant differences between viral variants.

## Supporting information

**S1 Table. Characteristic mutations of the Alpha variant.**
(DOCX)

**S2 Table. Full lineage characterisation of clinical severity dataset.**
(DOCX)

**S3 Table. Parameter estimates (on the linear predictor scale) from the severity model from the full dataset.**
(DOCX)

**S4 Table. Parameter estimates (on the linear predictor scale) from the severity model from the data subset excluding patients in nursing homes.**
(DOCX)

**S5 Table. Parameter estimates (on the linear predictor scale) from the severity model from the data subset only including hospitalised patients.**
(DOCX)

**S6 Table. Parameter estimates from the Ct value model.**
(DOCX)

**S1 Appendix. Further methods.**
(DOCX)

**S2 Appendix. Phylogenetic severity model.**
(DOCX)

**S3 Appendix. Comorbidities.**
(DOCX)

## Acknowledgments

We would like to thank all NHS staff that looked after patients during the COVID-19 pandemic in Scotland. The authors would like to acknowledge that this work uses data provided by patients and collected by the National Health Service (NHS) as part of their care and support. The authors would also like to acknowledge the work of the West of Scotland Safe Haven team in supporting extractions and linkage to de-identified NHS patient datasets. The authors would also like to acknowledge the work of the COG-UK consortium whose members are listed below:

Dr Samuel C Robson PhD [13, 84], Dr Thomas R Connor PhD [11, 74], Prof Nicholas J Loman PhD [43], Dr Tanya Golubchik PhD [5], Dr Rocio T Martinez Nunez PhD [46], Dr David Bonsall PhD [5], Prof Andrew Rambaut DPhil [104], Dr Luke B Snell MSc, MBBS [12], Rich Livett MSc [116], Dr Catherine Ludden PhD [20, 70], Dr Sally Corden PhD [74], Dr Eleni Nastouli FRCPath [96, 95, 30], Dr Gaia Nebbia PhD, FRCPath [12], Ian Johnston BSc [116], Prof Katrina Lythgoe PhD [5], Dr M. Estee Torok FRCP [19, 20], Prof Ian G Goodfellow PhD [24], Dr Jacqui A Prieto PhD [97, 82], Dr Kordo Saeed MD, FRCPath [97, 83], Dr David K Jackson PhD [116], Dr Catherine Houlihan PhD [96, 94], Dr Dan Frampton PhD [94, 95], Dr William L Hamilton PhD [19], Dr Adam A Witney PhD [41], Dr Giselda Bucca PhD [101], Dr Cassie F Pope PhD[40, 41], Dr Catherine Moore PhD [74], Prof Emma C Thomson PhD, FRCP [53], Dr Teresa Cutino-Moguel PhD [2], Dr Ewan M Harrison PhD [116, 102], Prof Colin P Smith PhD [101], Fiona Rogan BSc [77], Shaun M Beckwith MSc [6], Abigail Murray Degree [6], Dawn Singleton HNC [6], Dr Kirstine Eastick PhD, FRCPath [37], Dr Liz A Sheridan PhD [98], Paul Randell MSc, PgD [99], Dr Leigh M Jackson PhD [105], Dr Cristina V Ariani PhD [116], Dr Sónia Gonçalves PhD [116], Dr Derek J Fairley PhD [3, 77], Prof Matthew W Loose PhD [18], Joanne Watkins MSc [74], Dr Samuel Moses MD [25, 106], Dr Sam Nicholls PhD [43], Dr Matthew Bull PhD [74], Dr Roberto Amato PhD [116], Prof Darren L Smith PhD [36, 65, 66], Prof David M Aanensen PhD [14, 116], Dr Jeffrey C Barrett PhD [116], Dr Beatrix Kele PhD [2], Dr Dinesh Aggarwal MRCP[20, 116, 70], Dr James G Shepherd MBCHB, MRCP [53], Dr Martin D Curran

PhD [71], Dr Surendra Parmar PhD [71], Dr Matthew D Parker PhD [109], Dr Catryn Williams PhD [74], Dr Sharon Glaysher PhD [68], Dr Anthony P Underwood PhD [14, 116], Dr Matthew Bashton PhD [36, 65], Dr Nicole Pacchiarini PhD [74], Dr Katie F Loveson PhD [84], Matthew Byott MSc [95, 96], Dr Alessandro M Carabelli PhD [20], Dr Kate E Templeton PhD [56, 104], **Prof Sharon J Peacock PhD [20, 70]***, Dr Thushan I de Silva PhD [109], Dr Dennis Wang PhD [109], Dr Cordelia F Langford PhD [116], John Sillitoe BEng [116], Prof Rory N Gunson PhD, FRCPath [55], Dr Simon Cottrell PhD [74], Dr Justin O'Grady PhD [75, 103], Prof Dominic Kwiatkowski PhD [116, 108], Dr Patrick J Lillie PhD, FRCP [37], Dr Nicholas Cortes MBCHB [33], Dr Nathan Moore MBCHB [33], Dr Claire Thomas DPhil [33], Phillipa J Burns MSc, DipRCPath [37], Dr Tabitha W Mahungu FRCPath [80], Steven Liggett BSc [86], Angela H Beckett MSc [13, 81], Prof Matthew TG Holden PhD [73], Dr Lisa J Levett PhD [34], Dr Husam Osman PhD [70, 35], Dr Mohammed O Hassan-Ibrahim PhD, FRCPath [99], Dr David A Simpson PhD [77], Dr Meera Chand PhD [72], Prof Ravi K Gupta PhD [102], Prof Alistair C Darby PhD [107], Prof Steve Paterson PhD [107], Prof Oliver G Pybus DPhil [23], Dr Erik M Volz PhD [39], Prof Daniela de Angelis PhD [52], Prof David L Robertson PhD [53], Dr Andrew J Page PhD [75], Dr Inigo Martincorena PhD [116], Dr Louise Aigrain PhD [116], Dr Andrew R Bassett PhD [116], Dr Nick Wong DPhil, MRCP, FRCPath [50], Dr Yusri Taha MD, PhD [89], Michelle J Erkiert BA [99], Dr Michael H Spencer Chapman MBBS [116, 102], Dr Rebecca Dewar PhD [56], Martin P McHugh MSc [56, 111], Siddharth Mookerjee MPH [38, 57], Stephen Aplin [97], Matthew Harvey [97], Thea Sass [97], Dr Helen Umpleby FRCP [97], Helen Wheeler [97], Dr James P McKenna PhD [3], Dr Ben Warne MRCP [9], Joshua F Taylor MSc [22], Yasmin Chaudhry BSc [24], Rhys Izuagbe [24], Dr Aminu S Jahun PhD [24], Dr Gregory R Young PhD [36, 65], Dr Claire McMurray PhD [43], Dr Clare M McCann PhD [65, 66], Dr Andrew Nelson PhD [65, 66], Scott Elliott [68], Hannah Lowe MSc [25], Dr Anna Price PhD [11], Matthew R Crown BSc [65], Dr Sara Rey PhD [74], Dr Sunando Roy PhD [96], Dr Ben Temperton PhD [105], Dr Sharif Shaaban PhD [73], Dr Andrew R Hesketh PhD [101], Dr Kenneth G Laing PhD[41], Dr Irene M Monahan PhD [41], Dr Judith Heaney PhD [95, 96, 34], Dr Emanuela Pelosi FRCPath [97], Siona Silviera MSc [97], Dr Eleri Wilson-Davies MD, FRCPath [97], Dr Helen Fryer PhD [5], Dr Helen Adams PhD [4], Dr Louis du Plessis PhD [23], Dr Rob Johnson PhD [39], Dr William T Harvey PhD [53, 42], Dr Joseph Hughes PhD [53], Dr Richard J Orton PhD [53], Dr Lewis G Spurgin PhD [59], Dr Yann Bourgeois PhD [81], Dr Chris Ruis PhD [102], Áine O'Toole MSc [104], Marina Gourtovaia MSc [116], Dr Theo Sanderson PhD [116], Dr Christophe Fraser PhD [5], Dr Jonathan Edgeworth PhD, FRCPath [12], Prof Judith Breuer MD [96, 29], Dr Stephen L Michell PhD [105], Prof John A Todd PhD [115], Michaela John BSc [10], Dr David Buck PhD [115], Dr Kavitha Gajee MBBS, FRCPath [37], Dr Gemma L Kay PhD [75], David Heyburn [74], Dr Themoula Charalampous PhD [12, 46], Adela Alcolea-Medina [32, 112], Katie Kitchman BSc [37], Prof Alan McNally PhD [43, 93], David T Pritchard MSc, CSci [50], Dr Samir Dervisevic FRCPath [58], Dr Peter Muir PhD [70], Dr Esther Robinson PhD [70, 35], Dr Barry B Vipond PhD [70], Newara A Ramadan MSc, CSci, FIBMS [78], Dr Christopher Jeanes MBBS [90], Danni Weldon BSc [116], Jana Catalan MSc [118], Neil Jones MSc [118], Dr Ana da Silva Filipe PhD [53], Dr Chris Williams MBBS [74], Marc Fuchs BSc [77], Dr Julia Miskelly PhD [77], Dr Aaron R Jeffries PhD [105], Karen Oliver BSc [116], Dr Naomi R Park PhD [116], Amy Ash BSc [1], Cherian Koshy MSc, CSci, FIBMS [1], Magdalena Barrow [7], Dr Sarah L Buchan PhD [7], Dr Anna Mantzouratou PhD [7], Dr Gemma Clark PhD [15], Dr Christopher W Holmes PhD [16], Sharon Campbell MSc [17], Thomas Davis MSc [21], Ngee Keong Tan MSc [22], Dr Julianne R Brown PhD [29], Dr Kathryn A Harris PhD [29, 2], Stephen P Kidd MSc [33], Dr Paul R Grant PhD [34], Dr Li Xu-McCrae PhD [35], Dr Alison Cox PhD [38, 63], Pinglawathee Madona [38, 63], Dr Marcus Pond PhD [38, 63], Dr Paul A Randell MBBCh [38, 63], Karen T Withell FIBMS [48], Cheryl Williams MSc [51], Dr Clive Graham MD [60], Rebecca Denton-Smith BSc [62], Emma Swindells BSc [62], Robyn Turnbull BSc [62], Dr Tim J Sloan PhD [67], Dr Andrew Bosworth PhD [70, 35], Stephanie Hutchings [70], Hannah M Pymont MSc [70], Dr Anna Casey PhD [76], Dr Liz Ratcliffe PhD [76], Dr Christopher R Jones PhD

[79, 105], Dr Bridget A Knight PhD [79, 105], Dr Tanzina Haque PhD, FRCPath [80], Dr Jennifer Hart MRCP [80], Dr Dianne Irish-Tavares FRCPath [80], Eric Witele MSc [80], Craig Mower BA [86], Louisa K Watson DipHE [86], Jennifer Collins BSc [89], Gary Eltringham BSc [89], Dorian Crudgington [98], Ben Macklin [98], Prof Miren Iturriza-Gomara PhD [107], Dr Anita O Lucaci PhD [107], Dr Patrick C McClure PhD [113], Matthew Carlile BSc [18], Dr Nadine Holmes PhD [18], Dr Christopher Moore PhD [18], Dr Nathaniel Storey PhD [29], Dr Stefan Rooke PhD [73], Dr Gonzalo Yebra PhD [73], Dr Noel Craine DPhil [74], Malorie Perry MSc [74], Dr Nabil-Fareed Alikhan PhD [75], Dr Stephen Bridgett PhD [77], Kate F Cook MScR [84], Christopher Fearn MSc [84], Dr Salman Goudarzi PhD [84], Prof Ronan A Lyons MD [88], Dr Thomas Williams MD [104], Dr Sam T Haldenby PhD [107], Jillian Durham BSc [116], Dr Steven Leonard PhD [116], Robert M Davies MA (Cantab) [116], Dr Rahul Batra MD [12], Beth Blane BSc [20], Dr Moira J Spyer PhD [30, 95, 96], Perminder Smith MSc [32, 112], Mehmet Yavus [85, 109], Dr Rachel J Williams PhD [96], Dr Adhyana IK Mahanama MD [97], Dr Buddhini Samaraweera MD [97], Sophia T Girgis MSc [102], Samantha E Hansford CSci [109], Dr Angie Green PhD [115], Dr Charlotte Beaver PhD [116], Katherine L Bellis [116, 102], Matthew J Dorman [116], Sally Kay [116], Liam Prestwood [116], Dr Shavanthi Rajatileka PhD [116], Dr Joshua Quick PhD [43], Radoslaw Poplawski BSc [43], Dr Nicola Reynolds PhD [8], Andrew Mack MPhil [11], Dr Arthur Morriss PhD [11], Thomas Whalley BSc [11], Bindi Patel BSc [12], Dr Iliana Georgana PhD [24], Dr Myra Hosmillo PhD [24], Malte L Pinckert MPhil [24], Dr Joanne Stockton PhD [43], Dr John H Henderson PhD [65], Amy Hollis HND [65], Dr William Stanley PhD [65], Dr Wen C Yew PhD [65], Dr Richard Myers PhD [72], Dr Alicia Thornton PhD [72], Alexander Adams BSc [74], Tara Annett BSc [74], Dr Hibo Asad PhD [74], Alec Birchley MSc [74], Jason Coombes BSc [74], Johnathan M Evans MSc [74], Laia Fina [74], Bree Gatica-Wilcox MPhil [74], Lauren Gilbert [74], Lee Graham BSc [74], Jessica Hey BSc [74], Ember Hilvers MPH [74], Sophie Jones MSc [74], Hannah Jones [74], Sara Kumziene-Summerhayes MSc [74], Dr Caoimhe McKerr PhD [74], Jessica Powell BSc [74], Georgia Pugh [74], Sarah Taylor [74], Alexander J Trotter MRes [75], Charlotte A Williams BSc [96], Leanne M Kermack MSc [102], Benjamin H Foulkes MSc [109], Marta Gallis MSc [109], Hailey R Hornsby MSc [109], Stavroula F Louka MSc [109], Dr Manoj Pohare PhD [109], Paige Wolverson MSc [109], Peijun Zhang MSc [109], George MacIntyre-Cockett BSc [115], Amy Trebes MSc [115], Dr Robin J Moll PhD [116], Lynne Ferguson MSc [117], Dr Emily J Goldstein PhD [117], Dr Alasdair Maclean PhD [117], Dr Rachael Tomb PhD [117], Dr Igor Starinskij MSc, MRCP [53], Laura Thomson BSc [5], Joel Southgate MSc [11, 74], Dr Moritz UG Kraemer DPhil [23], Dr Jayna Raghwani PhD [23], Dr Alex E Zarebski PhD [23], Olivia Boyd MSc [39], Lily Geidelberg MSc [39], Dr Chris J Illingworth PhD [52], Dr Chris Jackson PhD [52], Dr David Pascall PhD [52], Dr Sreenu Vattipally PhD [53], Timothy M Freeman MPhil [109], Dr Sharon N Hsu PhD [109], Dr Benjamin B Lindsey MRCP [109], Dr Keith James PhD [116], Kevin Lewis [116], Gerry Tonkin-Hill [116], Dr Jaime M Tovar-Corona PhD [116], MacGregor Cox MSci [20], Dr Khalil Abudahab PhD [14, 116], Mirko Menegazzo [14], Ben EW Taylor MEng [14, 116], Dr Corin A Yeats PhD [14], Afrida Mukaddas BTech [53], Derek W Wright MSc [53], Dr Leonardo de Oliveira Martins PhD [75], Dr Rachel Colquhoun DPhil [104], Verity Hill [104], Dr Ben Jackson PhD [104], Dr JT McCrone PhD [104], Dr Nathan Medd PhD [104], Dr Emily Scher PhD [104], Jon-Paul Keatley [116], Dr Tanya Curran PhD [3], Dr Sian Morgan FRCPath [10], Prof Patrick Maxwell PhD [20], Prof Ken Smith PhD [20], Dr Sahar Eldirdiri MBBS, MSc, FRCPath [21], Anita Kenyon MSc [21], Prof Alison H Holmes MD [38, 57], Dr James R Price PhD [38, 57], Dr Tim Wyatt PhD [69], Dr Alison E Mather PhD [75], Dr Timofey Skvortsov PhD [77], Prof John A Hartley PhD [96], Prof Martyn Guest PhD [11], Dr Christine Kitchen PhD [11], Dr Ian Merrick PhD [11], Robert Munn BSc [11], Dr Beatrice Bertolusso Degree [33], Dr Jessica Lynch MBCHB [33], Dr Gabrielle Vernet MBBS [33], Stuart Kirk MSc [34], Dr Elizabeth Wastnedge MD [56], Dr Rachael Stanley PhD [58], Giles Idle [64], Dr Declan T Bradley PhD [69, 77], Nicholas F Killough MSc [69], Dr Jennifer Poyner MD [79], Matilde Mori BSc [110], Owen Jones BSc [11], Victoria Wright BSc [18], Ellena Brooks MA [20], Carol M Churcher BSc [20], Dr Laia Delgado Callico PhD [20],

Mireille Fragakis HND [20], Dr Katerina Galai PhD [20, 70], Dr Andrew Jermy PhD [20], Sarah Judges BA [20], Anna Markov BSc [20], Georgina M McManus BSc [20], Kim S Smith [20], Peter M D Thomas-McEwen MSc [20], Dr Elaine Westwick PhD [20], Dr Stephen W Attwood PhD [23], Dr Frances Bolt PhD [38, 57], Dr Alisha Davies PhD [74], Elen De Lacy MPH [74], Fatima Downing [74], Sue Edwards [74], Lizzie Meadows MA [75], Sarah Jeremiah MSc [97], Dr Nikki Smith PhD [109], Luke Foulser [116], Amita Patel BSc [12], Dr Louise Berry PhD [15], Dr Tim Boswell PhD [15], Dr Vicki M Fleming PhD [15], Dr Hannah C Howson-Wells PhD [15], Dr Amelia Joseph PhD [15], Manjinder Khakh [15], Dr Michelle M Lister PhD [15], Paul W Bird MSc, MRes [16], Karlie Fallon [16], Thomas Helmer [16], Dr Claire L McMurray PhD [16], Mina Odedra BSc [16], Jessica Shaw BSc [16], Dr Julian W Tang PhD [16], Nicholas J Willford MSc [16], Victoria Blakey BSc [17], Dr Veena Raviprakash MD [17], Nicola Sheriff BSc [17], Lesley-Anne Williams BSc [17], Theresa Feltwell MSc [20], Dr Luke Bedford PhD [26], Dr James S Cargill PhD [27], Warwick Hughes MSc [27], Dr Jonathan Moore MD [28], Susanne Stonehouse BSc [28], Laura Atkinson MSc [29], Jack CD Lee MSc [29], Dr Divya Shah PhD [29], Natasha Ohemeng-Kumi MSc [32, 112], John Ramble MSc [32, 112], Jasveen Sehmi MSc [32, 112], Dr Rebecca Williams BMBS [33], Wendy Chatterton MSc [34], Monika Pusok MSc [34], William Everson MSc [37], Anibolina Castigador IBMS HCPC [44], Emily Macnaughton FRCPath [44], Dr Kate El Bouzidi MRCP [45], Dr Temi Lampejo FRCPath [45], Dr Malur Sudhanva FRCPath [45], Cassie Breen BSc [47], Dr Graciela Sluga MD, MSc [48], Dr Shazaad SY Ahmad MSc [49, 70], Dr Ryan P George PhD [49], Dr Nicholas W Machin MSc [49, 70], Debbie Binns BSc [50], Victoria James BSc [50], Dr Rachel Blacow MBCHB [55], Dr Lindsay Coupland PhD [58], Dr Louise Smith PhD [59], Dr Edward Barton MD [60], Debra Padgett BSc [60], Garren Scott BSc [60], Dr Aidan Cross MBCHB [61], Dr Mariyam Mirfenderesky FRCPath [61], Jane Greenaway MSc [62], Kevin Cole [64], Phillip Clarke [67], Nichola Duckworth [67], Sarah Walsh [67], Kelly Bicknell [68], Robert Impey MSc [68], Dr Sarah Wyllie PhD [68], Richard Hopes [70], Dr Chloe Bishop PhD [72], Dr Vicki Chalker PhD [72], Dr Ian Harrison PhD [72], Laura Gifford MSc [74], Dr Zoltan Molnar PhD [77], Dr Cressida Auckland FRCPath [79], Dr Cariad Evans PhD [85, 109], Dr Kate Johnson PhD [85, 109], Dr David G Partridge FRCP, FRCPath [85, 109], Dr Mohammad Raza PhD [85, 109], Paul Baker MD [86], Prof Stephen Bonner PhD [86], Sarah Essex [86], Leanne J Murray [86], Andrew I Lawton MSc [87], Dr Shirelle Burton-Fanning MD [89], Dr Brendan AI Payne MD [89], Dr Sheila Waugh MD [89], Andrea N Gomes MSc [91], Maimuna Kimuli MSc [91], Darren R Murray MSc [91], Paula Ashfield MSc [92], Dr Donald Dobie MBCHB [92], Dr Fiona Ashford PhD [93], Dr Angus Best PhD [93], Dr Liam Crawford PhD [93], Dr Nicola Cumley PhD [93], Dr Megan Mayhew PhD [93], Dr Oliver Megram PhD [93], Dr Jeremy Mirza PhD [93], Dr Emma Moles-Garcia PhD [93], Dr Benita Percival PhD [93], Megan Driscoll BSc [96], Leah Ensell BSc [96], Dr Helen L Lowe PhD [96], Laurentiu Maftei BSc [96], Matteo Mondani MSc [96], Nicola J Chaloner BSc [99], Benjamin J Cogger BSc [99], Lisa J Easton MSc [99], Hannah Huckson BSc [99], Jonathan Lewis MSc, PgD, FIBMS [99], Sarah Lowdon BSc [99], Cassandra S Malone MSc [99], Florence Munemo BSc [99], Manasa Mutingwende MSc [99], Roberto Nicodemi BSc [99], Olga Podplomyk FD [99], Thomas Somassa BSc [99], Dr Andrew Beggs PhD [100], Dr Alex Richter PhD [100], Claire Cormie [102], Joana Dias MSc [102], Sally Forrest BSc [102], Dr Ellen E Higginson PhD [102], Mailis Maes MPhil [102], Jamie Young BSc [102], Dr Rose K Davidson PhD [103], Kathryn A Jackson MSc [107], Dr Alexander J Keeley MRCP [109], Prof Jonathan Ball PhD [113], Timothy Byaruhanga MSc [113], Dr Joseph G Chappell PhD [113], Jayasree Dey MSc [113], Jack D Hill MSc [113], Emily J Park MSc [113], Arezou Fanaie MSc [114], Rachel A Hilson MSc [114], Geraldine Yaze MSc [114], Stephanie Lo [116], Safiah Afifi BSc [10], Robert Beer BSc [10], Joshua Maksimovic FD [10], Kathryn McCluggage Masters [10], Karla Spellman FD [10], Catherine Bresner BSc [11], William Fuller BSc [11], Dr Angela Marchbank BSc [11], Trudy Workman HNC [11], Dr Ekaterina Shelest PhD [13, 81], Dr Johnny Debebe PhD [18], Dr Fei Sang PhD [18], Dr Sarah Francois PhD [23], Bernardo Gutierrez MSc [23], Dr Tetyana I Vasylyeva DPhil [23], Dr Flavia Flaviani PhD [31], Dr Manon Ragonnet-Cronin PhD [39], Dr Katherine L Smollett PhD [42], Alice Broos BSc [53], Daniel Mair

BSc [53], Jenna Nichols BSc [53], Dr Kyriaki Nomikou PhD [53], Dr Lily Tong PhD [53], Ioulia Tsatsani MSc [53], Prof Sarah O'Brien PhD [54], Prof Steven Rushton PhD [54], Dr Roy Sanderson PhD [54], Dr Jon Perkins MBCHB [55], Seb Cotton MSc [56], Abbie Gallagher BSc [56], Dr Elias Allara MD, PhD [70, 102], Clare Pearson MSc [70, 102], Dr David Bibby PhD [72], Dr Gavin Dabrera PhD [72], Dr Nicholas Ellaby PhD [72], Dr Eileen Gallagher PhD [72], Dr Jonathan Hubb PhD [72], Dr Angie Lackenby PhD [72], Dr David Lee PhD [72], Nikos Manesis [72], Dr Tamyo Mbisa PhD [72], Dr Steven Platt PhD [72], Katherine A Twohig [72], Dr Mari Morgan PhD [74], Alp Aydin MSci [75], David J Baker BEng [75], Dr Ebenezer Foster-Nyarko PhD [75], Dr Sophie J Prosolek PhD [75], Steven Rudder [75], Chris Baxter BSc [77], Sílvia F Carvalho MSc [77], Dr Deborah Lavin PhD [77], Dr Arun Mariappan PhD [77], Dr Clara Radulescu PhD [77], Dr Aditi Singh PhD [77], Miao Tang MD [77], Helen Morcrette BSc [79], Nadua Bayzid BSc [96], Marius Cotic MSc [96], Dr Carlos E Balcazar PhD [104], Dr Michael D Gallagher PhD [104], Dr Daniel Maloney PhD [104], Thomas D Stanton BSc [104], Dr Kathleen A Williamson PhD [104], Dr Robin Manley PhD [105], Michelle L Michelsen BSc [105], Dr Christine M Sambles PhD [105], Dr David J Studholme PhD [105], Joanna Warwick-Dugdale BSc [105], Richard Eccles MSc [107], Matthew Gemmell MSc [107], Dr Richard Gregory PhD [107], Dr Margaret Hughes PhD [107], Charlotte Nelson MSc [107], Dr Lucille Rainbow PhD [107], Dr Edith E Vamos PhD [107], Hermione J Webster BSc [107], Dr Mark Whitehead PhD [107], Claudia Wierzbicki BSc [107], Dr Adrienn Angyal PhD [109], Dr Luke R Green PhD [109], Dr Max Whiteley PhD [109], Emma Betteridge BSc [116], Dr Iraad F Bronner PhD [116], Ben W Farr BSc [116], Scott Goodwin MSc [116], Dr Stefanie V Lensing PhD [116], Shane A McCarthy [116, 102], Dr Michael A Quail PhD [116], Diana Rajan MSc [116], Dr Nicholas M Redshaw PhD [116], Carol Scott [116], Lesley Shirley MSc [116], Scott AJ Thurston BSc [116], Dr Will Rowe PhD[43], Amy Gaskin MSc [74], Dr Thanh Le-Viet PhD [75], James Bonfield BSc [116], Jennifer Liddle [116] and Andrew Whitwham BSc [116]

**1** Barking, Havering and Redbridge University Hospitals NHS Trust, **2** Barts Health NHS Trust, **3** Belfast Health & Social Care Trust, **4** Betsi Cadwaladr University Health Board, **5** Big Data Institute, Nuffield Department of Medicine, University of Oxford, **6** Blackpool Teaching Hospitals NHS Foundation Trust, **7** Bournemouth University, **8** Cambridge Stem Cell Institute, University of Cambridge, **9** Cambridge University Hospitals NHS Foundation Trust, **10** Cardiff and Vale University Health Board, **11** Cardiff University, **12** Centre for Clinical Infection and Diagnostics Research, Department of Infectious Diseases, Guy's and St Thomas' NHS Foundation Trust, **13** Centre for Enzyme Innovation, University of Portsmouth, **14** Centre for Genomic Pathogen Surveillance, University of Oxford, **15** Clinical Microbiology Department, Queens Medical Centre, Nottingham University Hospitals NHS Trust, **16** Clinical Microbiology, University Hospitals of Leicester NHS Trust, **17** County Durham and Darlington NHS Foundation Trust, **18** Deep Seq, School of Life Sciences, Queens Medical Centre, University of Nottingham, **19** Department of Infectious Diseases and Microbiology, Cambridge University Hospitals NHS Foundation Trust, **20** Department of Medicine, University of Cambridge, **21** Department of Microbiology, Kettering General Hospital, **22** Department of Microbiology, South West London Pathology, **23** Department of Zoology, University of Oxford, **24** Division of Virology, Department of Pathology, University of Cambridge, **25** East Kent Hospitals University NHS Foundation Trust, **26** East Suffolk and North Essex NHS Foundation Trust, **27** East Sussex Healthcare NHS Trust, **28** Gateshead Health NHS Foundation Trust, **29** Great Ormond Street Hospital for Children NHS Foundation Trust, **30** Great Ormond Street Institute of Child Health (GOS ICH), University College London (UCL), **31** Guy's and St. Thomas' Biomedical Research Centre, **32** Guy's and St. Thomas' NHS Foundation Trust, **33** Hampshire Hospitals NHS Foundation Trust, **34** Health Services Laboratories, **35** Heartlands Hospital, Birmingham, **36** Hub for Biotechnology in the Built Environment, Northumbria University, **37** Hull University Teaching Hospitals NHS Trust, **38** Imperial College Healthcare NHS Trust, **39** Imperial College London, **40** Infection Care Group, St George's University Hospitals NHS

Foundation Trust, **41** Institute for Infection and Immunity, St George's University of London, **42** Institute of Biodiversity, Animal Health & Comparative Medicine, **43** Institute of Microbiology and Infection, University of Birmingham, **44** Isle of Wight NHS Trust, **45** King's College Hospital NHS Foundation Trust, **46** King's College London, **47** Liverpool Clinical Laboratories, **48** Maidstone and Tunbridge Wells NHS Trust, **49** Manchester University NHS Foundation Trust, **50** Microbiology Department, Buckinghamshire Healthcare NHS Trust, **51** Microbiology, Royal Oldham Hospital, **52** MRC Biostatistics Unit, University of Cambridge, **53** MRC-University of Glasgow Centre for Virus Research, **54** Newcastle University, **55** NHS Greater Glasgow and Clyde, **56** NHS Lothian, **57** NIHR Health Protection Research Unit in HCAI and AMR, Imperial College London, **58** Norfolk and Norwich University Hospitals NHS Foundation Trust, **59** Norfolk County Council, **60** North Cumbria Integrated Care NHS Foundation Trust, **61** North Middlesex University Hospital NHS Trust, **62** North Tees and Hartlepool NHS Foundation Trust, **63** North West London Pathology, **64** Northumbria Healthcare NHS Foundation Trust, **65** Northumbria University, **66** NU-OMICS, Northumbria University, **67** Path Links, Northern Lincolnshire and Goole NHS Foundation Trust, **68** Portsmouth Hospitals University NHS Trust, **69** Public Health Agency, Northern Ireland, **70** Public Health England, **71** Public Health England, Cambridge, **72** Public Health England, Colindale, **73** Public Health Scotland, **74** Public Health Wales, **75** Quadram Institute Bioscience, **76** Queen Elizabeth Hospital, Birmingham, **77** Queen's University Belfast, **78** Royal Brompton and Harefield Hospitals, **79** Royal Devon and Exeter NHS Foundation Trust, **80** Royal Free London NHS Foundation Trust, **81** School of Biological Sciences, University of Portsmouth, **82** School of Health Sciences, University of Southampton, **83** School of Medicine, University of Southampton, **84** School of Pharmacy & Biomedical Sciences, University of Portsmouth, **85** Sheffield Teaching Hospitals NHS Foundation Trust, **86** South Tees Hospitals NHS Foundation Trust, **87** Southwest Pathology Services, **88** Swansea University, **89** The Newcastle upon Tyne Hospitals NHS Foundation Trust, **90** The Queen Elizabeth Hospital King's Lynn NHS Foundation Trust, **91** The Royal Marsden NHS Foundation Trust, **92** The Royal Wolverhampton NHS Trust, **93** Turnkey Laboratory, University of Birmingham, **94** University College London Division of Infection and Immunity, **95** University College London Hospital Advanced Pathogen Diagnostics Unit, **96** University College London Hospitals NHS Foundation Trust, **97** University Hospital Southampton NHS Foundation Trust, **98** University Hospitals Dorset NHS Foundation Trust, **99** University Hospitals Sussex NHS Foundation Trust, **100** University of Birmingham, **101** University of Brighton, **102** University of Cambridge, **103** University of East Anglia, **104** University of Edinburgh, **105** University of Exeter, **106** University of Kent, **107** University of Liverpool, **108** University of Oxford, **109** University of Sheffield, **110** University of Southampton, **111** University of St Andrews, **112** Viapath, Guy's and St Thomas' NHS Foundation Trust, and King's College Hospital NHS Foundation Trust, **113** Virology, School of Life Sciences, Queens Medical Centre, University of Nottingham, **114** Watford General Hospital, **115** Wellcome Centre for Human Genetics, Nuffield Department of Medicine, University of Oxford, **116** Wellcome Sanger Institute, **117** West of Scotland Specialist Virology Centre, NHS Greater Glasgow and Clyde, **118** Whittington Health NHS Trust

*Consortium lead–email: sjp97@medschl.cam.ac.uk

## Author Contributions

**Conceptualization:** Elen Vink, Rachel Blacow, Chris Davis, Ana da Silva Filipe, Guy Mollett, David L. Robertson, James G. Shepherd, Thomas Williams, Emma C. Thomson.

**Data curation:** David J. Pascall, Ana da Silva Filipe, Matthew T. G. Holden, Guy Mollett, Sharif Shabaan, James G. Shepherd, Emma C. Thomson.

**Formal analysis:** David J. Pascall, Elen Vink, Joseph Hughes, Samantha Lycett, James G. Shepherd.

**Funding acquisition:** Emma C. Thomson.

**Investigation:** Elen Vink, Rachel Blacow, Naomi Bulteel, Alasdair Campbell, Robyn Campbell, Sarah Clifford, Chris Davis, Ana da Silva Filipe, Noha El Sakka, Ludmila Fjodorova, Ruth Forrest, Emily Goldstein, Rory Gunson, Patrick Honour, Edward James, Tim Lewis, Martin McHugh, Guy Mollett, Yusuke Onishi, Ben Parcell, James G. Shepherd, Katherine Smollett, Kate Templeton, Elizabeth Wastnedge, Thomas Williams, Emma C. Thomson.

**Methodology:** David J. Pascall, Elen Vink, Chris Davis, Ana da Silva Filipe, Matthew T. G. Holden, Joseph Hughes, Samantha Lycett, Oscar MacLean, Guy Mollett, Surajit Ray, David L. Robertson, Sharif Shabaan, James G. Shepherd, Craig Wilkie, Thomas Williams, Emma C. Thomson.

**Project administration:** David J. Pascall, Elen Vink, John Haughney, Emma C. Thomson.

**Resources:** Emma C. Thomson.

**Supervision:** Emma C. Thomson.

**Visualization:** David J. Pascall, Joseph Hughes, Samantha Lycett, James G. Shepherd.

**Writing – original draft:** David J. Pascall, Guy Mollett, James G. Shepherd, Emma C. Thomson.

**Writing – review & editing:** David J. Pascall, Elen Vink, Rachel Blacow, Naomi Bulteel, Alasdair Campbell, Robyn Campbell, Sarah Clifford, Chris Davis, Ana da Silva Filipe, Noha El Sakka, Ludmila Fjodorova, Ruth Forrest, Emily Goldstein, Rory Gunson, John Haughney, Matthew T. G. Holden, Patrick Honour, Joseph Hughes, Edward James, Tim Lewis, Samantha Lycett, Oscar MacLean, Martin McHugh, Guy Mollett, Yusuke Onishi, Ben Parcell, Surajit Ray, David L. Robertson, Sharif Shabaan, James G. Shepherd, Katherine Smollett, Kate Templeton, Elizabeth Wastnedge, Craig Wilkie, Thomas Williams, Emma C. Thomson.

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
