## [Decision Letter · Decision Letter 0]

2 Feb 2023

PONE-D-22-22604The SARS-CoV-2 Alpha variant is associated with increased clinical severity of COVID-19 in Scotland: a genomics-based retrospective cohort analysisPLOS ONE

Dear Dr. Emma C. Thomson,

Thank you for submitting your manuscript to PLOS ONE. After careful consideration, we feel that it has merit but does not fully meet PLOS ONE’s publication criteria as it currently stands. Therefore, we invite you to submit a revised version of the manuscript that addresses the points raised during the review process.

We look forward to receiving your revised manuscript.

Kind regards,

Ali Amanati

Academic Editor

PLOS ONE

and https://journals.plos.org/plosone/s/file?id=ba62/PLOSOne_formatting_sample_title_authors_affiliations.pdf.

3. Thank you for stating the following in the Acknowledgments/ Funding Section of your manuscript:

“COG-UK is supported by funding from the Medical Research Council (MRC) part of UK Research & Innovation (UKRI), the National Institute of Health Research (NIHR) and Genome Research Limited, operating as the Wellcome Sanger Institute.  Funding was also provided by UKRI through the JUNIPER consortium (grant number MR/V038613/1).  Sequencing and bioinformatics support was funded by the Medical Research Council (MRC) core award (MC UU 1201412).”

“COG-UK is supported by funding from the Medical Research Council (MRC) part of UK Research & Innovation (UKRI), the National Institute of Health Research (NIHR) and Genome Research Limited, operating as the Wellcome Sanger Institute.  Funding was also provided by UKRI through the JUNIPER consortium (grant number MR/V038613/1).  Sequencing and bioinformatics support was funded by the Medical Research Council (MRC) core award (MC UU 1201412). The funders had no role in study design, data collection and analysis, decision to publish, or preparation of the manuscript.”

5. One of the noted authors is a group or consortium [The COVID-19 Genomics UK (COG-UK) consortium]. In addition to naming the author group, please list the individual authors and affiliations within this group in the acknowledgments section of your manuscript. Please also indicate clearly a lead author for this group along with a contact email address.

Additional Editor Comments:

Dear authors

Your manuscript [ID Number PONE-D-22-22604] has passed through the review stage and is ‎ready for revision. ‎

Editorial comments

To ensure the Editor and Reviewers can recommend that your revised manuscript is ‎accepted, ‎please pay careful attention to each of the comments posted underneath ‎this email. This way we ‎can avoid future rounds of clarifications and revisions, moving swiftly to ‎a decision.‎

‎1. Please provide a point-by-point response to the Editor and reviewer's comments

‎2. Please highlight all the amends on your manuscript with yellow color

‎3. Use line numbering and page number in the next submission‎

Other shortcomings

Considering the following statement in the method section: "Discharge status was followed up until 15th April 2021 for the hospital stay analysis.", Given the relatively long period of follow-up/observation, how did you control the possible impact of subsequent COVID-19 infections which may have occurred with different VOCs (other than the initial contracted infection) in the cases that were included early in the study (November 2020)?

"Nosocomial COVID-19 was defined as a first positive PCR occurring greater than 48 hours following admission to hospital.", How did you exclude Nosocomial infections? add more detail.

Regarding the "Clinical severity analysis" section, with the information provided, the readers will be expected to reach a better conclusion based on the models applied in the subgroup analysis. However, confusion remains, about whether another subgroup could be applied to the model?

The following statement seems incomplete: "Bernoulli models looking at sequential severity categories provided weak evidence that the proportional odds assumption of the cumulative logistic model was violated. The odds ratios for the no oxygen versus low flow oxygen, and low flow oxygen versus [high flow oxygen???] were ..."

You need to specify "worse outcomes" and "better outcomes" in the following statement (in this section and in the method): "A similar but more extreme effect was observed for the effect of biological sex, with male sex being associated with worse outcomes for the first two sequential category models (posterior median odds ratio for no oxygen versus low flow oxygen: 1.32, CI: 0.96-1.80; posterior median odds ratio for low flow oxygen versus high flow oxygen: 3.10, CI: 1.37-7.08), but better outcomes for the last (posterior odds ratio: 0.62, CI: 0.19-099)."

For the following statements remove the parentheses of confidence intervals: "posterior median Ct change: -2.46, 95% CI: (-4.22)-(-0.70)"; and "age and sex (HR: -0.02; 95% CI: (-0.23)-0.20; p = 0.89)."

Reviewers' comments:

Reviewer's Responses to Questions

**Comments to the Author**

1. Is the manuscript technically sound, and do the data support the conclusions?

Reviewer #1: Yes

Reviewer #2: Yes

Reviewer #3: Yes

2. Has the statistical analysis been performed appropriately and rigorously? 

Reviewer #1: Yes

Reviewer #2: Yes

Reviewer #3: Yes

3. Have the authors made all data underlying the findings in their manuscript fully available?

Reviewer #1: Yes

Reviewer #2: Yes

Reviewer #3: Yes

4. Is the manuscript presented in an intelligible fashion and written in standard English?

Reviewer #1: Yes

Reviewer #2: Yes

Reviewer #3: Yes

5. Review Comments to the Author

Reviewer #1: This is a really well written and well put together manuscript that addresses an important clinical area. I just have a couple of point which the authors could consider including either in their discussion or the methods in order to help thinking across organisations.

The spike gene drop out caused issues and delayed detection in some areas of the UK, was Scotland impacted by this and if so could the delay to diagnosis have played a role in increased disease severity?

The analysis does not take into account any immunosuppression or underlying conditions. Have the authors therefore assumed that the variants would have behaved in the same way across clinical groups and that they would have all received the same management options?

The authors stated that excluded nosocomial cases but it would be useful to know if they also excluded care home outbreaks in the analysis that include these patients as management is likely to have been different.

Healthcare worker cases appear to have been handled in the same analysis as others, would it have been worth undertaking a healthcare worker sub-analysis as they may have been exposed to multiple variants, higher loading doses and have been captured differently due to healthcare working testing provision?

Reviewer #2: The authors are to be commended for a well-conducted study. Some minor notes follow.

-Obviously, the alpha variant has all but disappeared and is primarily of historical note. The authors acknowledge this and suggest that this work may help in setting a baseline to which future variants may be compared. This is a reasonable suggestion, but if this is to be the case, absolute measures of risk including cumulative incidence estimates may be more helpful than relative measures. The authors conducted sound inferential work to determine measures of disease severity relative to previous (i.e., non-B.1.1.7) strains, but any absolute measures are difficult to find, and the reader is left scraping bits from Table 1 and Figure 2. I suggest foregrounding some absolute measures in the results and discussion to better serve the purposes suggested.

-Some of the typical potential confounding variables that spring to mind in a study like this-- age, calendar time (because of either changes in clinical care protocols or surges in hospital admissions), comorbidities-- were included as covariates in the primary model, as outlined in the supplementary material. I suggest discussing them directly, either in the Methods or Discussion, in order to forestall reader uncertainty.

-I wonder if the # of days since first diagnosis is an adequate control for potential hospital surges; there may have been variation in the timing of local outbreaks in the catchment area that left uncontrolled confounding.

-I believe the Fine and Gray method retains in the risk set those individuals who experienced a competing event not of interest (e.g., death, in the study of hospital stay length). Was the alpha variant associated with increased odds of death in this analysis?

-About 20% of samples were excluded because they could not be linked to patient records; is there a conceivable way that these could these have systematically differed from included samples by strain and outcomes (e.g., if they tended to be earlier samples among a low-income population that tends to have poorer health outcomes)?

Reviewer #3: Thanks so much for this submission and for conducting this important research. I greatly appreciate learning more about how genomic surveillance in Scotland is conducted, gaining insight on clinical severity of confirmed cases of the alpha variant, and your rationale on why it is important to study variants that are no longer dominant. With some minor revisions, I think this paper will contribute to both PLOS and existing science on COVID-19. Please see my suggested revisions by section below:

Introduction

• Optional: “The Alpha variant, the first variant of concern (VOC), was estimated to be 50-100% more transmissible than others present," would say other lineages rather than others present

• “A clear understanding of the behaviour of the Alpha variant, which emerged as a dominant variant, is needed as a baseline to compare the clinical phenotype of later variants, such as the currently dominant Omicron sub-variant BA.5.” Please cite a source stating dominant variant and verify that it is not BQ.1/BQ.1.1 or another variant when re-submitting since this is rapidly changing

Results

• Optional: For “At the time of the data used in this analysis” add a comma after this statement, consider re-wording to “at the time of data collection,”

• In table 1, is it possible to make “65.6±20. 6” and “67.2±20. 8” fit on one line? Right now, the 6 and 8 look like they are the range.

• For the other column in table 1, consider putting non-alpha in parentheses

• How do you define community transmission in Table 1? Consider adding in-text or as a footnote as I didn't see this.

• Optional: “1030 B.1.177 and 81 of 19 other lineage infections,” recommend rewording to say 81 cases from other lineages or something similar. I understand what you are saying, but from first glance it seemed like you were saying you had 81 of 19 lineages. Recommend to say 1030 B.1.177 cases and 81 cases of other lineage infections?

Discussion

• In the timeframe of this study, there was likely a very limited supply of vaccine available, particularly for healthcare workers. Were there any post-vaccination cases of COVID-19 in this study? Were there any exclusions made based on this, or is this something you don’t know? I suspect this could be a small source of bias if not accounted for. Consider mentioning vaccination status in discussion. If not knowing vaccination status is indeed a limitation, you could mention that post-vaccination cases were less likely in this time period due to the fact that most people had recently received their vaccine and the high vaccine efficacy at this time.

• “In summary, the Alpha variant was found to be associated with a rapid increase in SARS-CoV-2 cases”. Would say cases of COVID-19 instead of cases of SARS-CoV-2, since COVID-19 is the disease. If using SARS-CoV-2, SARS-CoV-2 is the virus that causes COVID-19.

6. PLOS authors have the option to publish the peer review history of their article (what does this mean?). If published, this will include your full peer review and any attached files.

Reviewer #1: **Yes: **Dr Elaine Cloutman-Green

Reviewer #2: No

Reviewer #3: No

---

## [Author Response · Author response to Decision Letter 0]

8 Mar 2023

A letter is enclosed with a point-by-point response to the reviewer and editor comments

---

## [Editor Report · Decision Letter 1]

27 Mar 2023

The SARS-CoV-2 Alpha variant was associated with increased clinical severity of COVID-19 in Scotland: a genomics-based retrospective cohort analysis

PONE-D-22-22604R1

Dear Dr. Emma C. Thomson,

We’re pleased to inform you that your manuscript has been judged scientifically suitable for publication and will be formally accepted for publication once it meets all outstanding technical requirements.

Kind regards,

Ali Amanati

Academic Editor

PLOS ONE

Additional Editor Comments (optional):

I read the revised manuscript ‎

I have no further comments to add. I thank the authors for their very detailed ‎‎replies to the reviewer and Editor's comments.‎

---

## [Editor Report · Acceptance letter]

3 Apr 2023

PONE-D-22-22604R1 

The SARS-CoV-2 Alpha variant was associated with increased clinical severity of COVID-19 in Scotland: a genomics-based retrospective cohort analysis 

Dear Dr. Thomson:

I'm pleased to inform you that your manuscript has been deemed suitable for publication in PLOS ONE. Congratulations! Your manuscript is now with our production department. 

Kind regards, 

on behalf of

Professor Ali Amanati 

Academic Editor

PLOS ONE